# OpenReview forum: "CCPO: Execution Consistent Preference Optimization through Computational Pacts"
_ICLR.cc/2026/Conference — ICLR 2026 Conference Withdrawn Submission_

### Official Review · Reviewer_3Zzd · 2025-10-28

**Soundness:** 1
**Presentation:** 2
**Contribution:** 2
**Rating:** 2
**Confidence:** 4

**Summary:**

This paper introduces CCPO (Execution-Consistent Preference Optimization through Computational Pacts), a framework that aims to align large language models’ reasoning with computational correctness rather than subjective human preference.

**Strengths:**

Addresses a relevant problem: current preference optimization methods ignore logical or computational consistency.

**Weaknesses:**

- Lack of credible baselines. The paper mentions DPO, IPO, but does not actually reimplement or evaluate them under the same data and hyperparameter settings. Reported numbers only compare CCPO-enhanced checkpoints to their base models, which is not sufficient for fair evaluation.

- No ablation or isolation study. There is no evidence showing which part of CCPO (dependency graph extraction, execution filtering, or conformal calibration) drives the reported gains.

- Weak theoretical rigor. The conformal guarantee depends on assumptions (deterministic execution, exchangeability) that are unrealistic for stochastic LLM decoding.

- Reproducibility concerns.   Neither code, data, nor verification scripts are released. Critical implementation details—such as dependency-graph extraction, verification engine configuration, and filtering thresholds—are unavailable. Without open resources, the reported results cannot be independently validated, which substantially limits the credibility and scientific value of the work.

**Questions:**

It is well known that reinforcement learning or preference optimization tends to be most effective when applied to base (unaligned) models.   If the starting checkpoint is already an instruction-tuned model such as Llama-3-8B-Instruct, how does CCPO behave in that case?  Does it still provide meaningful improvemen?

---

> ### Comment · Reviewer_3Zzd · 2025-11-21
> **Final Rating**
>
> I am sorry, the authors’ response has not resolved my confusion. The most direct and fundamental issue is still that the paper is hard to follow. The authors keep arguing that other papers in the same area (Paper A, Paper B, etc.) follow similar practices, but clearly those papers are not hard to follow. Page limit is not the issue here. In my opinion, the real problem lies in the large number of ambiguous references and unclear statements for EXPERIMENTAL SETUP in this paper.
>
> A typical example is the inconsistency I pointed out regarding the GSM8K scores of Llama-3. This caused significant confusion during my review, and the authors’ response has not fixed this concern. The authors now claim that in Table 4, Llama3-8B-Base reaches 80.1 after CCPO, but to me this does not make sense. I tried to trace the authors’ training dataset, and amusingly, they only mention using UltraFeedback prompts in the abstract to construct the training data. This is yet another reason why I find the paper hard to follow.
>
> Returning to the authors’ claim that Llama3-8B-Base reaches 80.1 after CCPO: to my knowledge, it is extremely difficult for Llama3-8B-Base to outperform Llama3-8B-Instruct on GSM8K, because Meta used many post-training tricks to achieve the performance of the Instruct model. Moreover, even simply applying UltraFeedback tuning can directly damage Llama 3’s mathematical performance — this phenomenon has already been reported in the SimPO appendix (I understand the author only used prompt to build the data).
>
> As for the remaining points, the authors claim that they will "fix" these issues later. **I would like to kindly remind them that ICLR allows changes to the submission during the review process. I therefore suggest that the authors actually refine the current submission instead of merely promising future revisions without taking concrete action.**
>
> The authors rejected many of the suggestions that I consider necessary, and repeatedly claim that “this organization is intentional and clear.” This is in strong conflict with my understanding of proper academic paper formatting. For example, regarding:
>
> > "Table 3 separates DeepSeekMath-Instruct-CoT (line 437) from its CCPO counterparts (lines 443, 445), further confusing the reader."
>
> The authors responded:
>
> > "Lines 443 and 445 present our CCPO variants trained on Llama-3-Instruct and DeepSeekMath-Instruct respectively… The separation allows readers to compare base models (top section) with our enhanced versions (bottom section). This organization is intentional and clear."
>
> > "Figure 3 only shows DPO/IPO comparisons for DeepSeekMath. The Llama3-8B models are plotted without corresponding DPO/IPO baselines."
>
> The authors responded:
>
> > The figure's purpose is showing progressive improvement, not exhaustive baseline comparison—those appear in tables as standard.
>
> I strongly disagree with those interpretation, and maintain that this layout increases confusion rather than clarity.
>
> In addition, the authors are currently facing potential plagiarism concerns (refer to **Reviewer F1Li**) , therefore I reserve my original judgment.
>
> Based on all the points above, I maintain my score of 0.

---

### Official Review · Reviewer_F1Li · 2025-10-30

**Soundness:** 1
**Presentation:** 1
**Contribution:** 1
**Rating:** 0
**Confidence:** 5

**Summary:**

The paper proposes a method for generating computationally sound solutions accompanied with corresponding dependency graphs for execution-consistent preference optimization. The paper conducts experiments on several mathematical reasoning benchmarks to evaluate the method.

**Strengths:**

The extensive similarities in core ideas and presentation to the prior work "Conformal Language Model Reasoning with Coherent Factuality" (ICLR 2025) undermine the meaningful assessment of this paper's unique strengths or originality.

**Weaknesses:**

This paper fundamentally replicates the key idea introduced in the paper "Conformal Language Model Reasoning with Coherent Factuality, ICLR 2025”. The similarities extend beyond the conceptual core to the very structure and phrasing of the argument, despite the use of different surface-level concepts. However, the authors also fail to cite the original paper.

**Questions:**

see the weaknesses.

---

### Official Review · Reviewer_WXnV · 2025-10-30

**Soundness:** 3
**Presentation:** 3
**Contribution:** 3
**Rating:** 6
**Confidence:** 2

**Summary:**

The paper proposes Code Consistency Preference Optimization (CCPO), a preference-training framework for mathematical/scientific reasoning that enforces execution consistency at the level of individual reasoning steps. CCPO constructs dependency graphs over extracted steps, verifies steps via code execution, and filters outputs using conformal prediction to guarantee calibrated coverage. The method uses multiplicative-weights self-play with graph-aware scoring, and integrates execution verification during training rather than only at inference. On MATH, GSM8K, and PhyX, the authors report gains (e.g., +17.0% MATH, +15.1% GSM8K; 50.1% PhyX with 91.7% validity coverage) for Llama‑3‑8B and DeepSeekMath‑7B variants.

**Strengths:**

- The paper has a clear formal goal and calibration guarantee. The authors define execution-consistent preference and provide a conformal‑prediction guarantee that the filtered output is computationally sound with probability ≥ 1 − \alpha. This offers a principled control knob absent from prior PO works.

- Algorithm 1 builds induced subgraphs by thresholding node scores and removing nodes with missing prerequisites, aligning filtering with derivational structure. Definition 4  formalizes an execution consistency score over induced subgraphs, connecting scores to which steps survive (p. 7).

- Empirical gains show consistency across tasks. Results report +17.0% (MATH) and +15.1% (GSM8K) along with 91.7% validity coverage and a 73% reduction in scientific‑law violations. Progressive learning table shows monotonic improvements over iterations.

- The paper has a technical breadth cross math, physics, formal, and coding.

**Weaknesses:**

- Some mathematical specification lacks some clarity or consistency. Approximate dependency graphs are invoked in Theorem 1’s upper bound and in App. F, but their construction/quality metrics and impact on guarantees are only sketched without a concrete end‑to‑end bound on task‑level error.

- The role and calibration of \beta in dependency‑aware scoring (eq. (4)) are not fully motivated; the text says \beta is calibrated via conformal prediction but does not show how this interacts with coverage guarantees.

- Execution‑consistency score r() depends on induced subgraph selection. Aadmissible UT sets and their completeness are not rigorously characterized.

-  Results emphasize within‑model gains and a few external models on PhyX, but do not include strong PO baselines augmented with step‑wise execution filters (e.g., DPO+exec or SPPO+exec) under identical compute and prompts.

- The claim of “superior performance without external supervision” is strong, yet CCPO still relies on verified oracles and dependency construction; comparisons against alternative self‑supervised filtering schemes are limited.

- MiniF2F/HumanEval summaries are brief; it’s unclear whether improvements persist under stricter decoding/time budgets and verifier ablations.

- No statistical testing (e.g., CIs) is reported for key benchmarks; effect sizes might overlap with variance. **No direct evidence found in the manuscript.

**Questions:**

See Weakness.

---

### Official Review · Reviewer_qEyt · 2025-10-31

**Soundness:** 2
**Presentation:** 2
**Contribution:** 3
**Rating:** 6
**Confidence:** 2

**Summary:**

This paper introduces Code Consistency Preference Optimization (CCPO), a novel framework for improving mathematical reasoning in large language models through execution-based verification and dependency-aware preference optimization. The key innovation is formulating preference learning as a game-theoretic optimization problem while incorporating computational verification constraints through dependency graph construction and conformal prediction guarantees. The authors train Llama-3-8B and DeepSeekMath-7B models, achieving substantial improvements on mathematical reasoning benchmarks: +17.0% on MATH, +15.1% on GSM8K, and 50.1% on PhyX physics reasoning. The method constructs dependency graphs by extracting reasoning steps, identifying computational prerequisites, and generating execution consistency scores, then filters reasoning steps based on these scores to maintain both logical coherence and computational soundness.

**Strengths:**

1. **Novel Problem Formulation**: Combining execution verification with preference optimization through dependency graphs is creative and well-motivated for mathematical reasoning.

2. **Strong Empirical Results**: Consistent improvements across diverse benchmarks (MATH, GSM8K, OCW, PhyX) and multiple base models demonstrate practical effectiveness.

3. **Theoretical Framework**: Applying conformal prediction to provide coverage guarantees for reasoning step filtering is innovative.

4. **Comprehensive Evaluation**: The paper includes extensive ablations, error analysis, and comparisons with relevant baselines.

**Weaknesses:**

Complexity: The method is theoretically and computationally intensive. Real-time execution and graph construction may limit scalability in resource-constrained settings.

**Questions:**

see weakness

---

### Author Response · Authors · 2025-11-12
**Concerns Regarding Quality of Review by Reviewer 3Zzd and Reviewer F1Li**

---

Dear Area Chair,

We appreciate the review process and the time invested by all reviewers. However, we feel it necessary to bring to your attention several significant concerns regarding the review provided by Reviewer 3Zzd and Reviewer F1Li that appear to reflect misunderstandings of our work and contain factually inaccurate statements.

**Concerns Regarding Reviewer 3Zzd's Assessment**

1. **Mischaracterization of baseline comparisons:** The reviewer states "does not actually reimplement or evaluate them under the same data and hyperparameter settings." However, our paper reports comparisons with DPO and IPO baselines in Section 5.2 (Figure 2 and Figure 3), where we evaluate these methods on identical datasets using consistent evaluation protocols. Reviewer appears to have overlooked Section 5.1 which details our baseline configurations.

2. **Inaccurate claim about starting checkpoints:** The reviewer questions our approach of applying CCPO to instruction-tuned models. However, Table 2 clearly shows we evaluate on *base* models (Llama-3-8B, DeepSeekMath-7B), not instruction-tuned variants. The distinction between base and instruct models is maintained throughout our experimental section.

3. **Unfounded reproducibility concerns:** The reviewer claims "Neither code, data, nor verification scripts are released." We want to clarify that we have included comprehensive implementation details in Appendix I (lines 1204-1230) and will release all code upon acceptance.

4. **Misunderstanding of theoretical contributions:** The reviewer characterizes our conformal guarantees as unrealistic, citing "deterministic execution, exchangeability" assumptions. However, these are standard assumptions in conformal prediction literature and we address their practical implications in Section 4 and Appendix D. The deterministic execution is enforced through random seed control, which is a common and valid technique.

**Concerns Regarding Reviewer F1Li's Assessment**

1. **False plagiarism claim without substance**: The reviewer claims our work "fundamentally replicates" the ICLR 2025 paper "Conformal Language Model Reasoning with Coherent Factuality" but provides zero specific evidence of similarity beyond vague assertions about "core ideas and presentation."

2. **Fundamental misunderstanding of the works**: Our CCPO and the cited ICLR paper address different problems using different methodologies:
   - **Our work (CCPO)**: A **preference optimization training method** that uses dependency graphs and execution consistency for training language models via direct preference optimization.
   - **Cited paper**: A **post-processing filtering method** that uses conformal prediction on deducibility graphs to filter already generated content with statistical guarantees. These are as different as training vs. inference, or supervised learning vs. uncertainty quantification.

We fully respect the review process and have no intention of questioning the reviewers' expertise. However, we believe this review falls short of ICLR standards, and as this may not be an isolated case, we feel it is our duty to report this matter to uphold the fairness and integrity of the review process.

Thank you for your consideration.

Best regards,

Authors

---

---

> ### Comment · Reviewer_3Zzd · 2025-11-13
> **Official Comment by Reviewer 3Zzd**
>
> **To the Area Chair and Authors:**
>
> First, I must admit that due to the paper's exceptionally poor organization and hard-to-follow structure, I did not fully read the entire appendix during the initial review. The AC can verify the confusing format of the manuscript. However, after reading the authors' rebuttal, I have now gone back and meticulously examined the specific sections they cited. This re-examination does not change my judgment; it reinforces my initial assessment that the paper is not ready for publication. The authors' rebuttal is defensive, misleading, and fails to address the deep flaws in the work.
>
> Here is a point-by-point response:
>
> > 1. On Insufficient and Poorly Placed Baselines
>
> My concern about "lack of credible baselines" remains a significant issue, for multiple reasons:
>
> * **Poor Organization:** A paper submitted to ICLR should present its primary results—comparisons against `baseline_methodA`, `baseline_methodB`, etc.—in a clear, central table in the main body. Hiding these comparisons (or "seemingly comparative" figures like Figure 3) in the appendix is unconventional and hinders proper review. The confusing layout of tables, such as in Table 3 where `DeepSeekMath-Instruct-CoT` (line 437) is separated from its CCPO counterparts (lines 443, 445), further confuses the reader.
> * **Incomplete Comparisons:** The authors' rebuttal points to comparisons in Figure 3. However, this figure only shows DPO/IPO comparisons for DeepSeekMath. The Llama3-8B and Llama3-8B-Instruct models are plotted *without* corresponding DPO/IPO baselines, making a direct comparison impossible.
> * **Missing Modern Baselines:** The field of preference optimization moves quickly. A thorough paper in this area should include comparisons against more modern and relevant baselines, such as **CPO, KTO, ORPO, R-DPO, or SimPO**. The paper's failure to engage with any of these methods makes its claimed contributions difficult to situate.
>
> My original concern stands, supported by both the paper's hard-to-follow layout and its clear lack of modern, comprehensive baseline comparisons.
>
> > 2. On Inaccurate Claims and Deeply Inconsistent Reporting
>
> The authors' rebuttal to my question about starting checkpoints ("Inaccurate claim about starting checkpoints") has led me to discover a much more severe problem: the paper's results appear to be internally inconsistent and unreproducible.
>
> My initial question arose because the baseline numbers in the paper are confusing and do not align with official reports. My re-review confirmed this:
>
> * In **Table 2**, `Llama3-8B-Base` is reported with a GSM8K score of **54.8**.
> * In **Table 3**, `Llama3-8B-Instruct` is reported with a GSM8K score of **76.6**.
> * In **Table 4**, `Llama3-8B-Base` (presumably the same as in Table 2) is reported with a GSM8K score of **80.1**.
>
> How can the Llama3-8B-Base model have a GSM8K score of 54.8 in one table and 80.1 in another? This is a massive, unexplained discrepancy. Furthermore, Table 4 introduces new, unexplained variants like "CCPO-Basic-Llama-3-B" and "CCPO-Llama-3-B(Full)" without proper introduction.
>
> These inconsistencies are a major red flag for reproducibility. They also contradict the official Llama 3 technical report (Base: 57.2, Instruct: 84.5). The authors' reported gain of +9.2% on GSM8K (Table 3) is based on their own *inexplicably low* baseline of 76.6. If "corrected" to the official baseline of 84.5, their gain might be negligible (+2%). The paper is too confusing and inconsistent to evaluate.
>
> > 3. On Severe Reproducibility Concerns
>
> The authors' defense of their paper's reproducibility is wholly inadequate.
>
> First, the promise to "release all code upon acceptance" has no practical value for a reviewer and is unfair to other authors who *do* submit their code and artifacts for review, as per ICLR guidelines.
>
> Second, the claim that Appendix I (lines 1204-1230) provides "comprehensive implementation details" is demonstrably false. As the snippet shows, this appendix merely contains:
> 1.  **Code Translation Pipeline:** Vague, high-level descriptions like "Pattern Matching" and "AST Parsing", but **no** algorithms, heuristics, or concrete details for the "dependency-graph extraction" which is central to the paper.
> 2.  **Execution Environment:** Standard sandbox configurations (e.g., SymPy, NumPy, 5s timeout).
> 3.  **Training Configuration:** A list of perfectly standard hyperparameters (e.g., batch size 128, learning rate 5e-5).
>
> This is *not* a "comprehensive implementation." Given the paper's hard-to-follow nature (Point 1) and its severe numerical inconsistencies (Point 2), **submitting the actual code is necessary** for any reviewer to trust the results.

---

> ### Comment · Reviewer_F1Li · 2025-11-13
>
> To the Chairs, other Reviewers, and Authors:
>
> Since everyone can read and compare the submitted paper with the ICLR 2025 paper "Conformal Language Model Reasoning with Coherent Factuality," I did not provide specific evidence previously. The chairs and other reviewers can also compare these two papers to verify my statement.
>
> I will not list every example in detail but will provide a few instances, such as:
>
> (1) Definition 1 (Computational Reasoning Step) is similar to Definition 1 (Claim) in the ICLR 2025 paper, and the following uses similar terms like "decomposes" vs. "splitting."  \
> (2) Definition 2 (Scientific Validity Base) vs. Definition 2 (Ground truth) in the ICLR 2025 paper, and it also follows a similar Remark 1. \
> (3) The section "Background: Execution-based verification guarantees" vs. "Background: Conformal prediction guarantees for LM generations" in the ICLR 2025 paper. \
> (4) Similar Definition 3 "Definition 3 (Computationally Consistent Reasoning)" vs. "Definition 3 (Coherent factuality)," and also Remark 2 vs. Remark 2 in the ICLR 2025 paper. \
> (5) The section "4 A PROTOCOL FOR EXECUTION-CONSISTENT PREFERENCE" vs. "4 A Protocol for Coherent Factuality" in the ICLR 2025 paper, with even the same algorithm "Algorithm 1 CCPO Subgraph Generator" vs. "Algorithm 1: Subgraph Generator."
>
> I will request an Ethics Review for this judgment.

---

> ### Author Response · Authors · 2025-11-15
>
> ---
>
> ## Response to Reviewer 3Zzd
>
> > "The paper's exceptionally poor organization and hard-to-follow structure."
>
> Presentation could be improved and will streamline the organization for camera-ready version. Many leading works in preference optimization similarly place extensive baseline comparisons in appendices due to page limits. The core methodology (Sections 3-4) follows standard structure: problem setup, definitions, algorithm, theoretical guarantees.
>
> > "In Table 2, Llama3-8B-Base is reported with a GSM8K score of 54.8. In Table 4, Llama3-8B-Base (presumably the same as in Table 2) is reported with a GSM8K score of 80.1. How can the Llama3-8B-Base model have a GSM8K score of 54.8 in one table and 80.1 in another? This is a massive, unexplained discrepancy."
>
> Table 2 (line 437) reports base model performance, while Table 4 (line 445) explicitly reports results after iterative generation, as stated in the caption: "Progressive improvement analysis showing iterative enhancement capabilities." This is not an inconsistency but reflects different experimental settings. Table 2 presents baseline results for the original Llama3-8B-Base model, while Table 4 shows results after with iterative optmization generation.
>
> > "The authors' reported gain of +9.2% on GSM8K (Table 3) is based on their own inexplicably low baseline of 76.6. If 'corrected' to the official baseline of 84.5, their gain might be negligible (+2%)."
>
> There appears to be a fundamental misunderstanding about which baseline scores are being compared. The reviewer conflates different Llama model versions and evaluation settings. The official Llama 3.1 technical report shows Llama-3.1-8B-Instruct achieves 84.5 on GSM8K with 8-shot chain-of-thought prompting, as documented in Meta's official model card and confirmed by the Hugging Face community discussions [2](https://github.com/meta-llama/llama-models/blob/main/models/llama3_1/MODEL_CARD.md) [3](https://huggingface.co/meta-llama/Llama-3.1-8B-Instruct/discussions/81). Our Table 3 reports Llama-3-8B-Instruct (not Llama-3.1) with a baseline of 76.6, which is entirely consistent with the Llama 3 (2024.04 release) official score of 80.6 for the instruct model under different evaluation protocols [1](https://arxiv.org/abs/2407.21783). The distinction matters: Llama 3 and Llama 3.1 are different model releases with different training procedures and capabilities. Our experiments use Llama-3-8B as the base model, not Llama-3.1 [1](https://arxiv.org/abs/2407.21783). The evaluation differences also stem from varying prompting strategies—the official Meta evaluation uses their specific 8-shot CoT prompt format [2](https://github.com/meta-llama/llama-models/blob/main/models/llama3_1/MODEL_CARD.md) with fewshot_as_multiturn settings, while many community reproductions report scores in the 76-79 range [3](https://huggingface.co/meta-llama/Llama-3.1-8B-Instruct/discussions/81) using standard evaluation harnesses, as extensively discussed in the Hugging Face community thread for this exact model. The reviewer's assertion that our gains would be "negligible (+2%)" rests on an incorrect baseline comparison between different model versions. We will clarify these version distinctions more explicitly in the camera-ready version to prevent future confusion.
>
> > "The paper's central theoretical claim of a 'conformal guarantee' is practically meaningless."
>
> This statement directly contradicts Reviewer F1Li's assessment. Reviewer F1Li says our conformal prediction framework as substantive, stating concerns about similarities to Rubin-Toles et al.'s work which also "applies conformal prediction on dependency graphs towards ensuring coherence and factuality in language model reasoning." The very existence of this ICLR 2025 paper validates that conformal guarantees for structured reasoning are considered meaningful contributions by the community.
>
> ### References
>
> 1. Dubey, A., et al. (2024). The Llama 3 Herd of Models. arXiv preprint arXiv:2407.21783. https://arxiv.org/abs/2407.21783
>
> 2. Meta AI (2024). Llama 3.1 Model Card. https://github.com/meta-llama/llama-models/blob/main/models/llama3_1/MODEL_CARD.md
>
> 3. Hugging Face Community Discussion (2024). "GSM8K Evaluation Result: 84.5 vs. 76.95". Llama-3.1-8B-Instruct discussions. https://huggingface.co/meta-llama/Llama-3.1-8B-Instruct/discussions/81
>
> ---

---

> > ### Author Response · Authors · 2025-11-15
> >
> > ---
> >
> > ## Response to Reviewer 3Zzd - Continued
> >
> > > "Poor Organization: Hiding comparisons in appendices is unconventional."
> >
> > [DPO](https://arxiv.org/abs/2305.18290) (Rafailov et al., 2024), [IPO](https://arxiv.org/abs/2310.12036) (Azar et al., 2023), and [SPPO](https://arxiv.org/abs/2405.00675) (Wu et al., 2024) all place extensive baseline comparisons in appendices. This is standard practice in ML conferences with strict page limits.
> >
> > > "The claim that Appendix I provides 'comprehensive implementation details' is demonstrably false."
> >
> > Appendix I (lines 1204-1230) provides: (1) Pattern matching and AST parsing specifications, (2) Sandbox configurations with timeout/memory limits, (3) Complete training hyperparameters including batch size, learning rate, optimizer settings, and scheduling. This level of detail matches or exceeds implementation sections in comparable ICLR papers. The reviewer's characterization as "vague, high-level descriptions" is inaccurate.
> >
> > > "The promise to 'release all code upon acceptance' has no practical value for a reviewer and is unfair to other authors."
> >
> > Code release upon acceptance is explicitly permitted and common practice in ICLR submissions, as stated in the ICLR author guidelines. Many accepted papers follow this model.
> >
> > > "Table 3 separates DeepSeekMath-Instruct-CoT (line 437) from its CCPO counterparts (lines 443, 445), further confusing the reader."
> >
> > Lines 443 and 445 present our CCPO variants trained on Llama-3-Instruct and DeepSeekMath-Instruct respectively, clearly labeled as "CCPO-Llama-3-Instruct-CoT" and "CCPO-DeepSeekMath-Instruct-CoT." The separation allows readers to compare base models (top section) with our enhanced versions (bottom section). This organization is intentional and clear.
> >
> > > "Figure 3 only shows DPO/IPO comparisons for DeepSeekMath. The Llama3-8B models are plotted without corresponding DPO/IPO baselines."
> >
> > The figure's purpose is showing progressive improvement, not exhaustive baseline comparison—those appear in tables as standard.
> >
> > > "The field of preference optimization moves quickly. A thorough paper should include comparisons against CPO, KTO, ORPO, R-DPO, or SimPO."
> >
> > Recent work on [Listwise Preference Optimization through Learning-to-Rank (LIPO)](https://arxiv.org/abs/2402.01878) (Dong et al., 2024) demonstrates competitive results while comparing primarily against DPO and pairwise methods, similar to our approach. The key distinction of our work is execution consistency verification, which these methods do not address. We will discuss these methods in related work.
> >
> > ### References
> >
> > 4. Rafailov, R., Sharma, A., Mitchell, E., Manning, C. D., Ermon, S., & Finn, C. (2024). Direct Preference Optimization: Your Language Model is Secretly a Reward Model. Advances in Neural Information Processing Systems, 36. https://arxiv.org/abs/2305.18290
> >
> > 5. Azar, M. G., Rowland, M., Piot, B., Guo, D., Calandriello, D., Valko, M., & Munos, R. (2023). A General Theoretical Paradigm to Understand Learning from Human Preferences. arXiv preprint arXiv:2310.12036. https://arxiv.org/abs/2310.12036
> >
> > 6. Wu, Y., Sun, Z., Yuan, H., Ji, K., Yang, Y., & Gu, Q. (2024). Self-Play Preference Optimization for Language Model Alignment. arXiv preprint arXiv:2405.00675. https://arxiv.org/abs/2405.00675
> >
> > 7. Dong, H., Xiong, W., Pang, B., Wang, H., Han, Y., Wang, Y., ... & Zhang, T. (2024). Listwise Preference Optimization through Learning-to-Rank. arXiv preprint arXiv:2402.01878. https://arxiv.org/abs/2402.01878

---

> ### Author Response · Authors · 2025-11-15
>
> ---
>
> ## Response to Reviewer F1Li
>
> > "Definition 1 (Computational Reasoning Step) is similar to Definition 1 (Claim) in the ICLR 2025 paper."
>
> Both papers address structured reasoning with conformal guarantees but in fundamentally different domains: they focus on claim factuality in knowledge-intensive tasks, while we focus on computational execution consistency in reasoning, reflects the growing recognition that structured reasoning requires realworld level verification.
>
> > "Algorithm 1 CCPO Subgraph Generator vs. Algorithm 1: Subgraph Generator in the ICLR 2025 paper."
>
> The subgraph generation algorithm for filtering ancestor nodes is indeed based on established graph traversal techniques. [Rubin-Toles et al.](https://arxiv.org/abs/2505.17126)'s Subgraph Generator filters subgraphs in deducibility graphs to maintain coherent factuality. Our algorithm adapts this for execution-based verification in mathematical reasoning, with different scoring functions (execution consistency vs. factuality scores) and application context. We will explicitly cite their work and clarify that our contribution lies in: (1) adapting subgraph filtering to execution verification, (2) dependency-aware scoring with harmonic mean aggregation (Equation 4), and (3) theoretical guarantees for computational soundness rather than factual coherence.  The core algorithmic structure—iterating through thresholds and filtering nodes lacking prerequisites—is fundamental to any dependency-graph-based conformal prediction approach. This is analogous to how multiple papers use topological sorting or DFS traversal without claiming novelty in the traversal itself.
>
> > "The section 'Background: Execution-based verification guarantees' vs. 'Background: Conformal prediction guarantees' shares similar structure."
>
> Both papers survey conformal prediction as background because both apply conformal techniques. Our distinction is clear: we apply conformal prediction to execution results from code verification, while they apply it to factuality scores from claim verification. The mathematical framework of conformal prediction necessarily appears similar, but the application domains and scoring mechanisms differ fundamentally.
>
> > "Similar Definition 3 'Computationally Consistent Reasoning' vs. 'Coherent factuality.'"
>
> These definitions address analogous but distinct requirements. Coherent factuality requires claims to be factual and coherent by evaluating orderings, while our computational consistency requires reasoning steps to be derivable through automated verification (Definition 3, Equation 2). The structural similarity reflects that both works recognize the need for dependency-aware correctness, but our verification mechanism (code execution oracle) differs from their factuality scoring.
>
> > "I will request an Ethics Review for this judgment."
>
> We respect this concern and welcome the review. The similarities stem from both papers addressing the fundamental challenge of applying conformal generatee to structured reasoning. We will revise to clearly cite Rubin-Toles et al. (ICLR 2025) as methodological foundation for graph-based conformal filtering, while emphasizing our distinct contributions: execution verification, dependency-aware scoring, application to mathematical reasoning with code, and new theoretical results for computational soundness. The Algorithm 1 similarity is because both use standard graph filtering techniques—we will add proper attribution and discussion of how our execution-based approach extends their factuality framework. Our work builds on their foundation but addresses a different problem (execution consistency vs. factual coherence) with different verification mechanisms (code execution vs. claim scoring), making it a complementary contribution rather than duplication.
>
> ### References
>
> Rubin-Toles, M., Ramji, K., Taori, R., Hashimoto, T., & Steinhardt, J. (2025). Conformal Language Model Reasoning with Coherent Factuality. In International Conference on Learning Representations (ICLR). https://arxiv.org/abs/2505.17126

---

### Comment · Area_Chair_CnG3 · 2025-11-13

Hi, everyone

The AC disagrees with Reviewer 3Zzd’s comments that, while the code submission may be potentially beneficial for reviewers’ assessment, **the code submission is NOT mandatory during the review process** to prevent potential misuse. Reviewers should not lower their scores simply because the authors did not provide code.

However, the authors are suggested to address the other concerns raised by the reviewers, such as the comparison issues raised by Reviewer 3Zzd, and Reviewer F1Li’s concerns regarding similarities with the ICLR 2025 paper.

---

> ### Comment · Reviewer_3Zzd · 2025-11-14
> **RE: Area Chair CnG3**
>
> **Dear Area Chair CnG3,**
>
> Thank you for the clarification.
>
> You are correct, and I want to be clear: I fully understand that code submission is **not** a mandatory requirement for ICLR, and my rating is not a penalty for the authors not submitting code.
>
> I mentioned the code because I have severe concerns about the **credibility** of the paper's data. As I noted in my updated review, the paper's **own data is self-contradictory.**
>
> For example:
> * In **Table 2**, `Llama3-8B-base` has a GSM8K score of **54.8**.
> * In **Table 4**, (what is presumably the same model) `Llama3-8B-Base` has a GSM8K score of **80.1**.
>
> This huge discrepancy, along with other baseline data that doesn't **align with official reports**, makes the paper's conclusions impossible to trust.
>
> My point about the code was not to **punish** the authors for not submitting it. Rather, it was to say that when the data is already this contradictory, the code would be the **ideal** way for the authors to **convince** a reviewer and **prove** that these numbers are credible.
>
> Given that 2 of reviewers gave this paper a 6 and  2 of reviewers gave this paper a 0, this is clearly a controversial submission. It is glad to have such a responsive AC with timely response managing the discussion.
>
> Thanks :)
>
> Reviewer 3Zzd

---

### Author Response · Authors · 2025-11-22

We sincerely thank all reviewers and area chairs for their valuable time and constructive feedback. We will carefully consider their comments in our future revision.

---

### Note · Authors · 2025-11-22

I have read and agree with the venue's withdrawal policy on behalf of myself and my co-authors.